https://doi.org/10.1038/s41467-020-14692-4　　OPEN

# A square-root topological insulator with non-quantized indices realized with photonic Aharonov-Bohm cages

Mark Kremer [1,3], Ioannis Petrides[2,3], Eric Meyer [1], Matthias Heinrich[1], Oded Zilberberg [2✉] & Alexander Szameit [1✉]

Topological Insulators are a novel state of matter where spectral bands are characterized by quantized topological invariants. This unique quantized nonlocal property commonly manifests through exotic bulk phenomena and corresponding robust boundary effects. In our work we study a system where the spectral bands are associated with non-quantized indices, but nevertheless possess robust boundary states. We present a theoretical analysis, where we show that the square of the Hamiltonian exhibits quantized indices. The findings are experimentally demonstrated by using photonic Aharonov-Bohm cages.

[1] Institut für Physik, Universität Rostock, Albert-Einstein-Straße 23, 18059 Rostock, Germany. [2] Institut für Theoretische Physik, ETH Zürich, Wolfgang-Pauli-Straße 27, 8093 Zürich, Switzerland. [3] These authors contributed equally: Mark Kremer, Ioannis Petrides. ✉email: odedz@phys.ethz.ch; alexander.szameit@uni-rostock.de

The Klein–Gordon Hamiltoninan is a famous example, where taking its square-root lead to fundamentally new insides: with the help of the resulting Dirac Hamiltonian, describing a massive spin-1/2 fermionic particle, it was possible to explain the fine-structure spectra of atoms and the anomalous Zeeman effect[1]. Interestingly, as it emerged much later, the square-root procedure has proven to be useful also in other fields: it explains the robust boundary modes of mechanical lattices and connects it to the topological band theory of electronic systems[2], it relates bosons and fermions via supersymmetric transformations[3], and, in addition, it generates rich models from nontrivial topological insulators (TIs)[4].

TIs—a new phase of matter—have to date seen a variety of manifestations with prominent examples including the two dimensional (2D)[5] and four dimensional (4D) quantum Hall effects[6], one dimensional (1D) topological superconductors[7], 2D[8] and three dimensional (3D) TIs[9], crystalline and quasi-crystalline TIs[10,11], and higher-order TIs[12]. All available realizations of TIs, however, share a common feature: their spectral bands are attributed a nonlocal topological index that is quantized[13–15]. Hence, whereas different realizations of TIs can vary locally, as long as their topological characterization persists, they will exhibit the same topological phenomena. In other words, the quantization of topological indices lies at the foundation of the characteristic robustness of bulk responses and associated boundary phenomena in TIs.

In our work, we use a square-root procedure to provide a topological framework of a 1D TI with non-quantized bulk indices. This explains the robust boundary states found in its spectrum. Specifically, we analyze a system with three spectral bands, which possess non-quantized Zak's phases. However, spectral symmetries lead to quantized topological invariants, revealed when squaring the Hamiltonian, which determine its topological phase. The resulting states, related to the invariants by the bulk boundary correspondence, may be seen as in-gap, protected and controllable qubits. Photonic platforms have proven to serve as versatile platforms for the implementation of topological phenomena[15], such as Floquet TIs[16], TIs on a silicon platform[17], 2D[11] and 4D topological Hall physics[18], as well as non-Hermitian topological physics[19]. Along these lines, we utilize photonic waveguide arrays with a specifically tailored effective negative hopping to implement our theoretical findings.

## Results

**Theoretical model.** We consider a chain made of Aharonov–Bohm cages, i.e., a quasi-1D lattice composed of interconnected plaquettes, see Fig. 1a. Each lattice site is coupled to its neighbors with hopping amplitude $t$, while each plaquette is threaded by a flux $\phi$. The momentum space Hamiltonian of this model is given by

$$H(k) = t \begin{pmatrix} 0 & 1+e^{-ik} & e^{-i\phi}+e^{-ik} \\ 1+e^{ik} & 0 & 0 \\ e^{i\phi}+e^{ik} & 0 & 0 \end{pmatrix} \equiv t \sum_{i=1}^{4} d_i \lambda_{3+i},$$

(1)

where $\lambda_i$, with $i = 1, \ldots, 8$, are the eight Gell–Mann matrices (defined in Supplementary Note 5) and $\mathbf{d} = (d_1, d_2, d_3, d_4)$ is a 4-component real-valued vector with $d_1 = 1 + \cos k$, $d_2 = \sin k$, $d_3 = \cos\phi + \cos k$, and $d_4 = \sin\phi + \sin k$. The spectrum of $H(k)$ has three bands: a central band that remains nondispersive for all values of the flux $\phi$, and two additional particle–hole symmetric bands. For $\phi = 0$, the three bands cross, while for $\phi = \pi$ the spectrum is gapped with three flat bands at energies $E_i \in \{-2t, 0, 2t\}$, see

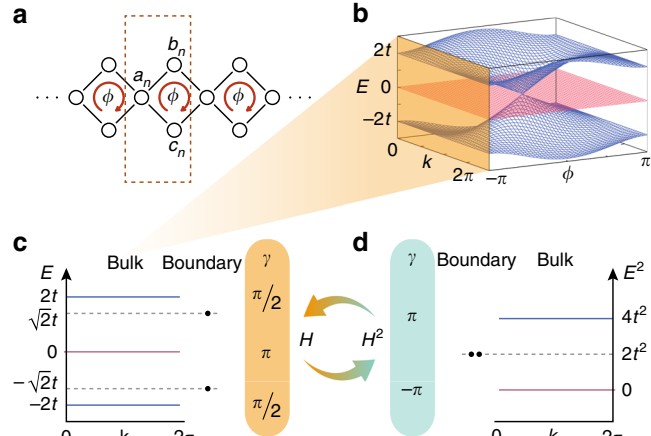

**Fig. 1 The Aharonov–Bohm cages. a** A chain of Aharonov–Bohm cages [cf. Eq. (1)], with three sites $a_n$, $b_n$, $c_n$ in the $n$th unit cell and a flux $\phi$ threading each plaquette. **b** The energy dispersion $E(k)$ of the chain as a function of the flux $\phi$. **c** The energy dispersion $E(k)$ at $\phi = \pi$ consists of three flat band at energies 0 and $\pm 2t$. The band at $E = 0$ has a quantized Zak's phase $\gamma = \pi$ while the other two bands show a non-quantized winding of $\pi/2$. At a termination of the chain with site $a_n$, two in-gap boundary states appear at $E = \pm\sqrt{2}t$. **d** Squaring the Hamiltonian (1) yields a model (2) with one flat band at $E = 0$ and two degenerate flat bands at $E = 4t^2$. Both bands have a quantized Wilzcek–Zee phase $|\gamma| = \pi$.

Fig. 1b. The latter case corresponds to the Aharonov–Bohm caging effect, where the particles become immobile due to destructive interference[20–22].

For each band, we can evaluate a 1D topological invariant, Zak's winding phase $\gamma_i = \int_{BZ} dk \mathcal{A}_i(k)$, where $\mathcal{A}_i(k) = i\langle v_i(k)|\partial_k|v_i(k)\rangle$ is the Berry connection of the $i$th band and $|v_i(k)\rangle$ is the corresponding eigenstate[23]. For a standard 1D TI (e.g., the Su–Shrieffer–Heeger (SSH) model), the winding phase takes quantized values of $\pi$ (or 0) corresponding to encircling (or not encircling) a singularity in quasi-momentum phase space[24]. For our model, we find that the zero-energy band has a winding phase $\gamma_2 = \pi$, whereas the top and bottom bands have $\gamma_1 = \gamma_3 = -\pi/2$. Moreover, the winding phases are quantized to these values, $\gamma_2 \in \{0, \pi\}$ mod $2\pi$ and $\gamma_1 = \gamma_3 \in \{0, \frac{\pi}{2}\}$ mod $2\pi$, by a nonsymmorphic transformation $\chi = \frac{1}{3}\mathbb{1} - e^{ik}\lambda_3 - \frac{1}{3}\lambda_8$ (see Supplementary Note 1 for the transformation for a general $\phi$ flux). The Hamiltonian $H(k)$ holds one additional non-symmorphic symmetry $\Pi = \frac{1}{3}\mathbb{1} + e^{ik}\lambda_3 - \frac{1}{3}\lambda_8$ (see Supplementary Note 1 for the transformation for a general $\phi$ flux) that quantizes the winding phases to $\gamma_2 \in \{0, \pi\}$ mod $2\pi$ and $\gamma_1 + \gamma_3 \in \{0, \pi\}$ mod $2\pi$[25]. Thus, a $\chi$-breaking term in the Hamiltonian $H(k)$ makes the phases $\gamma_1$ and $\gamma_3$ of the spectrally separated bands non-quantized and continuously mixed.

The AB-cages chain, with $\phi = \pi$ and open boundary conditions, has two in-gap states at energies $\pm\sqrt{2}t$ localized on the same boundary. Interestingly, their localization and energy are robust against disorder that does not break the $\Pi$- or $\chi$-symmetry (see Fig. 2 and Supplementary Note 7). Hence, the AB-cages chain has robust boundary states, even when the winding phases $\gamma_1$ and $\gamma_3$ are not quantized. Commonly, robust boundary states appear in a gap that lies above bands that have a quantized topological index[13–15]. The appearance of such symmetry-protected states is a surprising occurrence for our case where non-quantized bulk windings arise.

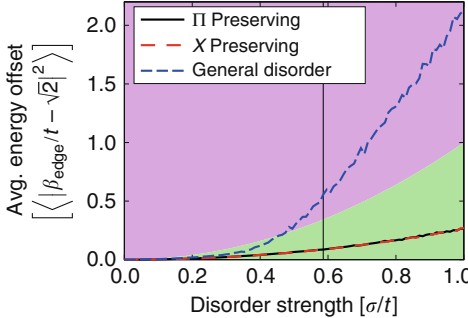

**Fig. 2 Disorder analysis.** The averaged mean squared difference $\langle|\beta_{\text{edge}}/t - \sqrt{2}|^2\rangle$ between the boundary state energy $\beta_{\text{edge}}$ in the presence of disorder and the energy of the undisturbed system $\beta_0 = \sqrt{2}t$, as a function of the disorder strength $\sigma$. Three different disorder types are used, which either do not break one of the symmetries ($\chi$ or $\Pi$) or break both. All three disorder types are chosen such that on average both symmetries, $\chi$ and $\Pi$, are preserved, i.e., the disorder distribution has a vanishing mean. The vertical solid line indicates the size of the gap between the boundary state and the nearest bulk band. The red and green regions define an energy offset that is bigger (red) or smaller (green) than $(\sigma/t)^2$, which corresponds to the energy scale of the disorder. The disorder-averaging simulations were run using a lattice with 99 sites and every disorder strength $\sigma$ was realized 10,000 times.

The topological aspects of the model are revealed by taking the square of the Hamiltonian matrix (1)

$$H^2(k) = t^2 \begin{pmatrix} 2m_0 & 0 & 0 \\ 0 & m_0 + m_3 & m_1 + im_2 \\ 0 & m_1 - im_2 & m_0 - m_3 \end{pmatrix} \qquad (2)$$

$$\equiv t^2 m_0 \left( \frac{4}{3}\mathbb{1} + \frac{1}{3}\lambda_8 \right) + t^2 \sum_{i=1}^{3} m_i \lambda_i,$$

where $m_0 = \sqrt{m_1^2 + m_2^2 + m_3^2}$, $m_1 = 1 + \cos(k) + \cos(\phi) + \cos(k - \phi)$, $m_2 = \sin(k) + \sin(\phi) - \sin(k - \phi)$, and $m_3 = \cos(k) - \cos(k - \phi)$. The squared Hamiltonian is block diagonal with a single band $|w_1\rangle$ at energy $\Lambda_1 = 2t^2 m_0$ and a $2 \times 2$ subblock with $|w_2\rangle$ and $|w_3\rangle$ eigenstates at energies $\Lambda_2 = 0$ and $\Lambda_3 = 2t^2 m_0$, respectively. The latter two form a subblock that corresponds to a topologically nontrivial 1D model which maps to the SSH model by a rotation with $e^{i\lambda_3 \frac{(\pi - \phi)}{4}} e^{i\lambda_2 \frac{\pi}{4}}$. Specifically, at $\phi = \pi$, the resulting $2 \times 2$ subblock is equivalent to the SSH-chain with 0 intra-cell coupling, $2t^2$ inter-cell coupling, and a constant $2t^2$ energy shift (see Supplementary Fig. 3).

Importantly, $|w_1\rangle$ and $|w_3\rangle$ form a degenerate subspace at energy $\Lambda_1 = \Lambda_3 = 2t^2 m_0$. Therefore, these bands are assigned a Wilczek–Zee phase which generalizes Zak's phase to multiband scenarios[26], $\gamma = \int_{\text{BZ}} \text{Tr}\,(\mathcal{A}(k)) dk$, where $\mathcal{A}(k)^{nm} = \langle v_n(k)|\partial_k|v_m(k)\rangle$, and $n$, $m$ run over the involved states. For the squared Hamiltonian $H^2(k)$, the Wilczek–Zee phase of both the zero-energy band and the degenerate subspace is quantized to $\{0, \pi\} \mod 2\pi$ due to the $\Pi$- and $\chi$-transformations (see Supplementary Note 3). As a result, the standard bulk-boundary correspondence of 1D TIs applies[27] and the open boundary spectrum of the Hamiltonian (2) maintains mid-gap states localized at the boundary. When $\phi = \pi$, the energy of these states is pinned to $2t^2$. Hence, the two energetically separated states appearing at the boundary of the AB cages are mapped, under the squaring operation, onto topological boundary states of the squared Hamiltonian. This leads to their characteristic robustness, both in localization and energy, against disorder that preserves the corresponding symmetry that quantizes the topological phases in $H^2(k)$ (see Fig. 2).

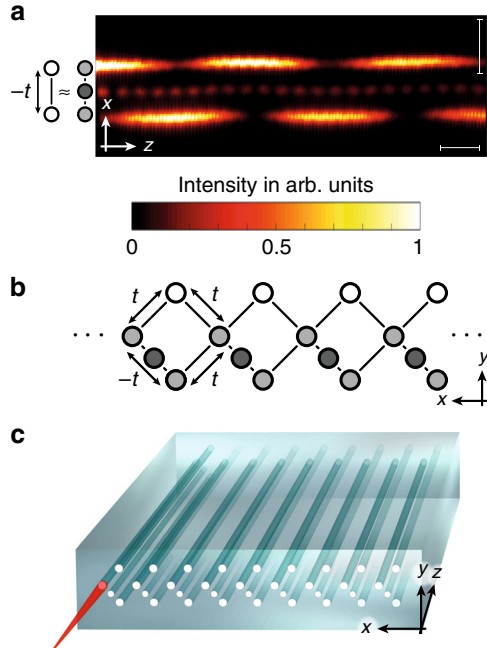

**Fig. 3 Experimental implementation. a** Light dynamics in the three level system of two waveguides and an auxiliary waveguide with carefully chosen refractive index. Since most of the amplitude is in the original waveguides, we can trace out the dynamics of the auxiliary waveguide and obtain a two-level system with an effective negative hopping $-t$. The horizontal scale bar corresponds to 1 cm, while the vertical scale bar corresponds to 25 μm. **b** Placing such a defect within each plaquette of the lattice structure generates a total flux of $\phi = \pi$. **c** An illustration of the quasi-1D array of evanescently coupled waveguides used in the experiment. Light is selectively injected into an input facet of the device and directly imaged using fluorescence microscopy.

**Experimental realization.** We implement the AB-cages chain (1) in photonic waveguide lattices fabricated using the femtosecond laser writing technique in bulk glass[28]. The evolution of light propagating along the $z$-direction of an array of single-mode waveguides can be well described in the paraxial approximation through a set of coupled mode equations $i\partial_z \psi = H\psi$. The wave-function $\psi$ represents the excited optical wavepacket as a super-position of bound modes of the waveguides. The matrix $H$ has diagonal elements corresponding to the refractive indices of the waveguides and off-diagonal coupling elements being proportional to the overlap between the bound modes of neighboring waveguides. Thus, discrete Schrödinger equations can be simulated in waveguide arrays with the benefit that the time coordinate in the quantum regime is mapped onto a spatial propagation distance in the optical system. In other words, the propagation of an optical wavepacket through a waveguide system simulates the temporal dynamics of an electron in a potential landscape. Notably, using fluorescence microscopy we can directly image the light propagation along the device[29].

In order to generate an effective AB-phase threading each cage, we use Peierls' substitution and associate an effective phase to one of the hopping amplitudes, see Fig. 3. Engineering a hopping phase for photons is challenging since the positive refractive index of each waveguide always results in a real and positive coupling between the waveguides. Nevertheless, by positioning an auxiliary waveguide with a well-tuned refractive index in between two waveguides[30], an effective negative coupling between the two original waveguides is generated (see Supplementary Note 8). Crucially, the auxiliary waveguide is engineered such that it does

not contribute significantly to the dynamics of the system, see Fig. 3a. By choosing the refractive index of the two original waveguides to energetically match the effective two-level system with the rest of the lattice (see Supplementary Note 8) and placing a negative coupling in each plaquette of our waveguide structure (see Fig. 3b), an overall flux of $\pi$ within the plaquettes is created, resulting in the desired AB-caging effect (see Fig. 3c for an illustration of the device).

We first establish our ability to generate the AB-caging effect in the bulk of the chain by probing the light dynamics to test the flatness of the bands, see Fig. 4. Exciting a single waveguide within a plaquette will excite all $k$-states of Bloch bands that overlap with this site. For flat bands, the light will stay bound to the injection point and will not disperse. We perform two experiments corresponding to two different injection sites within the unit cell, see Fig. 4a–c. Indeed, despite of some residual spreading due to imperfect injection and weak disorder in the device, in both experiments the propagating wavepacket remains confined to the injected unit cell. The experimental measurements agree well with tight-binding simulations of the AB chain. In contrast, light propagation for the case of vanishing flux $\phi = 0$ shows no localization and the wavepacket spreads to the entire lattice (see Supplementary Fig. 7).

From the light propagation along the sample (see Fig. 4), we can additionally measure the energy of the bands: launching light into a waveguide that connects two plaquettes solely excites the two states in the bands at $E = \pm 2t$, as the state from the band at $E = 0$ has no weight in this site, see Fig. 4c, d. The resulting beating pattern is, therefore, generated by two modes with a beating length $l_b$ that is connected to the energy difference $\Delta E$ of the participating modes by[31] $l_b = \frac{\pi}{\Delta E}$. From the beating in Fig. 4c, we measure $l_b = 0.9$ cm, which corresponds to $\Delta E = \pm 3.4$ cm$^{-1}$. Taking into account the particle-hole symmetry of the model, the energy of the two bands are therefore measured to be at $E = \pm 1.7$ cm$^{-1}$ while the third band lies at $E = 0$.

We, now, demonstrate the existence of the boundary states in our square-root model. The amplitude distribution of the predicted boundary modes is shown in Fig. 5a, b. The two states differ by a phase flip and appear at two inequivalent eigenenergies, cf. Fig. 1c. Hence, similarly to the bulk experiments above, light injected into the outermost waveguide simultaneously excites both boundary modes and the resulting light pattern exhibits a beating with a frequency corresponding to the difference between their eigenenergies, see Fig. 5c. Our experimental data agree well with tight-binding simulations shown in Fig. 5d. From the beating structure, we can determine the energy of the boundary modes $E_e$: we observe a beating with $l_b = 1.3$ cm and, hence, deduce that $E_e = \pm 1.2$ cm$^{-1}$. Comparing the observed energies in the bulk and in the boundary, we find that $\frac{E^2}{E_e^2} \sim 2$, in agreement with the predictions of the model (1).

## Discussion

In our work, we have predicted and demonstrated the physics of a square-root TI, using a photonic platform. Specifically, we show that the AB cages with $\phi = \pi$ have in-gap states at energies $\pm\sqrt{2}t$, above bands possessing $\pi/2$ or $\pi$ mod $2\pi$ Zak's phases. We find that these states are robust, both in energy and localization, against disorder that does not break the symmetries that quantize the topological indices in the corresponding system where the square of the Hamiltonian is taken. Furthermore, we show that the squaring operation bijectively maps the boundary states of $H$ to specific boundaries of $H^2$. Extending this description to the regime where $\gamma_1$ and $\gamma_3$ are not quantized, e.g., by adding a term that preserves $\Pi$ but breaks $\chi$, we find that the localization and energy deviation of the boundary states remain robust against

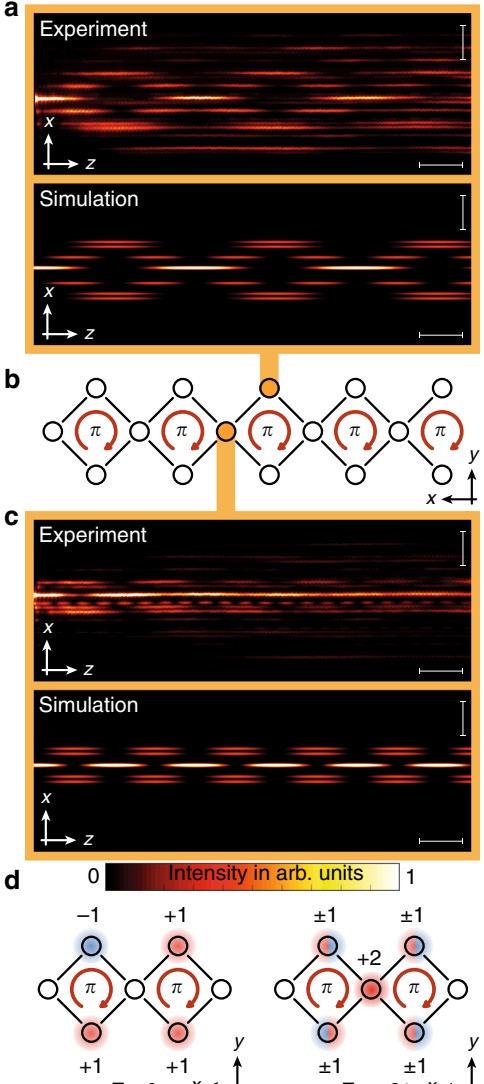

**Fig. 4 Bulk dynamics. a** Light dynamics when exciting the top waveguide in a bulk plaquette. The total envelope remains localized and shows breathing only within the plaquette. Differences between simulations and experiment arise mainly due to small amounts of light leaking into neighboring waveguides during the excitation of the waveguide. This generates a slightly different initial state launched into the system resulting in small deviations of the propagation dynamics. The horizontal scale bar corresponds to 1 cm, while the vertical scale bar corresponds to 50 μm. **b** The two waveguides that are probed in the experiments demonstrating the flatness of the bulk spectrum. **c** Light dynamics in the structure when a waveguide between two plaquettes is excited. The total envelope shows a local breathing while being localized within the plaquette. The horizontal scale bar corresponds to 1 cm, while the vertical scale bar corresponds to 50 μm. **d** The amplitude distribution of the three bulk eigenstates of the system.

additional $\Pi$-preserving disorder, but their energies are now fixed to a different value (see Supplementary Note 6). In this case, the mapping of the square operation is generalized using the properties of SU(3) algebra, while the symmetry protection in the square-root model still manifests due to the quantization of the topological indices of the squared Hamiltonian by the $\Pi$-symmetry (see Supplementary Note 6). Contrary to previous implementations of photonic AB cages using different experimental techniques[22], we provide an interpretation to the underlying physics which is independent of the specific model.

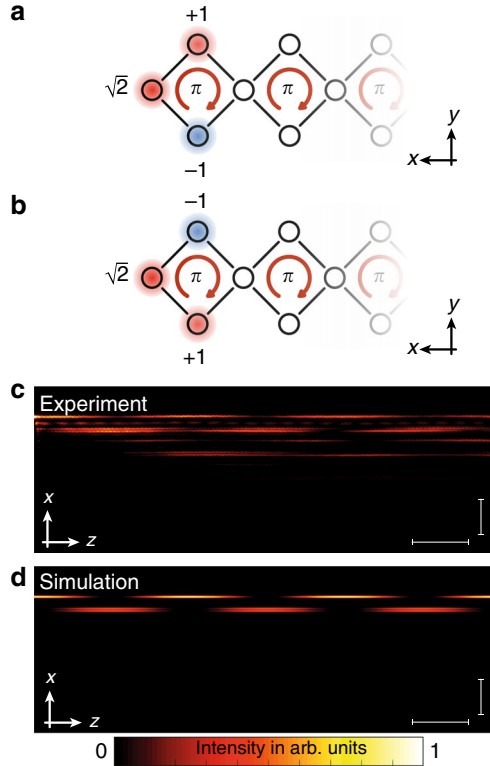

**Fig. 5 Boundary dynamics and symmetry protection. a** Amplitude distribution of the boundary state with energy $E = \sqrt{2}t$. **b** Amplitude distribution of the boundary state with energy $E = -\sqrt{2}t$. **c** Beating pattern between two boundary states excited by launching light into the outermost waveguide of the structure. The horizontal scale bar corresponds to 1 cm, while the vertical scale bar corresponds to 50 μm. **d** Tight-binding simulations confirming the behavior.

With our work, we hope to stimulate a range of new theoretical and experimental studies exploring the implications and breadth of such phases of matter. In this vein, our experimental results give rise to various important questions: First, can our square-root TI phase be realized in ultracold atomic setups, where topological quantities can be observed via bulk wavepacket dynamics, rather than by detection of boundary states? Second, the boundary states appearing at $\pm\sqrt{2}t$ energies form a two-level system (with states $1/2\left[\sqrt{2}|a\rangle \pm (|b\rangle - |c\rangle)\right]$) that can be used as a qubit that is energetically separated from the bulk bands and is robust to disorder. Making the hopping of the last site weaker or methodically breaking the nonsymmorphic symmetries of the AB-cages model offer a control handle to reduce or asymmetrically tune the energy splitting of the boundary states. Can this be used for single-qubit operations? Third, our procedure differs from recent works[2,4] where the square of a even-dimensional model has been used to map between models with different degrees of freedom but same quantized winding phase. In our work the dimension of the Hilbert space is fixed, and the topological invariant reveals itself upon squaring; the question arises if there are other nonlinear maps between Hamiltonians that admit such a description? Fourth, the phenomenology of our model resembles the valence-bond structure of the Affleck, Kennedy, Lieb, and Tasaki ground state of the Haldane spin-chain[32]: each unit cell in the bulk has three states that form a spin-1 subspace, coupling to their neighbors with a specific tunnel coupling such that an unpaired spin-1/2 is left at the boundary. Can our work suggest a connection to the topology of spin models? Fifth, the quantized $\pi/2$ phases of the AB-cages model in the electronic

domain will result into a boundary state with a $e/4$ charge. Can such a novel quasiparticle have nontrivial braiding statistics? Could its charge be tuned by controllably breaking the symmetries of the system? Finally, can the theoretical framework be generalized to higher dimensions? The answers to these questions are now in experimental reach.

## Methods

**Sample fabrication.** The waveguides were written inside a high-purity 10 cm long fused silica wafer (Corning 7980) using a RegA 9000 seeded by a Mira Ti:Al$_2$O$_3$ femtosecond laser. Pulses centered at 800nm with duration of 150 fs were used at a repetition rate of 100 kHz and energy of 450 nJ. The pulses were focused 500 μm under the sample surface using an objective with a numerical aperture of 0.35 while the sample was translated at constant speed of 40, 200, and 220 mm/min, corresponding to the different detunings, by high-precision positioning stages (ALS130, Aerotech Inc.). The mode field diameters of the guided mode were 10.4 μm × 8.0 μm at 633 nm. Propagation losses were estimated to be 0.2 dB/cm. The waveguides are equally spaced by 21.5 μm for the positive and 28 μm for the negative coupling, resulting in an inter-site hopping of $|t| = 0.85\,\text{cm}^{-1}$.

**Fluorescence imaging.** For the direct monitoring of the light propagation in our samples, we used a fluorescence microscopy technique[29]. A massive formation of nonbridging oxygen hole color centers occurs during the writing process, when fused silica with a high content of hydroxide is used, resulting in a homogeneous distribution of these color centers along the waveguides. When light from a Helium–Neon laser at $\lambda = 633$ nm is launched into the waveguides, the nonbridging oxygen hole color centers are excited and the resulting fluorescence ($\lambda = 650$ nm) can be directly observed using a CCD camera with an appropriate narrow linewidth filter. As the color centers are formed exclusively inside the waveguides, this technique yields a high signal-to-noise ratio.

## Data availability

The data that support the findings of this study are available from the corresponding author upon reasonable request.

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

## Acknowledgements
We thank J. Shapiro, G.-M. Graf, D. Bercioux, and J. Vidal for fruitful discussions. I.P. and O.Z. acknowledge financial support from the Swiss National Science Foundation (SNF). A.S. thanks the Deutsche Research Foundation (grants SZ 276/7-1, SZ 276/9-1, BL 574/13-1, and SZ 276/19-1) and the Krupp von Bohlen and Halbach foundation. The authors would also like to thank C. Otto for preparing the high-quality fused silica samples used in all experiments presented here.

## Author contributions
M.K., I.P., A.S., and O.Z. developed the concept, I.P. and O.Z. developed the theory, M.K., E.M., M.H., and A.S. designed the lattice structure, M.K. and E.M. performed the experiments, all authors discussed the results and co-wrote the paper.

## Competing interests
The authors declare no competing interests.
