## [Peer Review File · Nature Communications]

Editorial Note: This manuscript has been previously reviewed at another journal that is not operating a transparent peer review scheme. This document only contains reviewer comments and rebuttal letters for versions considered at Nature Communications .

Reviewers' comments:

Reviewer #2 (Remarks to the Author):

This work provides an experimental realization of a one-dimensional photonic lattice with non-integer Zak phases. In contrast to common sense that a topological insulator is usually associated with a quantized topological number, here the authors give an interesting new type of insulators with non-quantized indices yet robust boundary states. Interestingly, such a system returns to the SSH model after taken square of the Hamiltonian and applied a rotation. This thus explains the topological origin of the robustness of the boundary states. The boundary states and flat bands feature are experimentally verified and agree quite well with the theoretical predictions. The manuscript is well-written and the results are clearly presented. The physics discussed here is novel and quite interesting and I will recommend this work for publication provided the following questions have been well addressed.

1. A general question. Can this square-root topological insulator be extended to higher dimensional systems beside 1D? And can it be extended to be beyond the tight-binding model?
2. The introduction of this work focus on the meaning of square-root topological insulators and their importance. As pioneer and earlier works along this thought, the contributions of Refs. [37, 38] should be highlighted in the beginning.
3. At $\phi=0$, three bands cross. This feature cannot be observed in Fig. 1b. I guess a transparent mesh would be better.
4. The last paragraph on page 2, the authors mentioned that "For a standard TI, the winding phase takes ...". I guess here the authors refer to SSH model. As nowadays, the term TI include many different systems. I think it is better to be more precise here.
5. The authors adopt the method presented in Ref. [34] to introduce π flux. The authors should also mention that this method also introduce an onsite energy shift besides the negative hopping phase. What is the effect of this onsite energy shift?
6. Following this question, can arbitrary value of flux be realized in this system?
7. A few typos. Figs. 3 and 4, I can hardly believe the scale bar along the x direction is 50m.
8. The authors mentioned that the light field is measured directly through fluorescence microscopy. Does this mean that light is leakage from side.
9. A control experiment which shows how the wave propagate if the band is not flat would be great in supplement of Fig. 3.

Reviewer #2 (Remarks to the Author):**Comment:**

This work provides an experimental realization of a one-dimensional photonic lattice with non-integer Zak phases. In contrast to common sense that a topological insulator is usually associated with a quantized topological number, here the authors give an interesting new type of insulators with non-quantized indices yet robust boundary states. Interestingly, such a system returns to the SSH model after taken square of the Hamiltonian and applied a rotation. This thus explains the topological origin of the robustness of the boundary states. The boundary states and flat bands feature are experimentally verified and agree quite well with the theoretical predictions. The manuscript is well-written and the results are clearly presented. The physics discussed here is novel and quite interesting and I will recommend this work for publication provided the following questions have been well addressed.

Response:

We thank the reviewer for his/her time and effort, as well as the careful evaluation of our manuscript. We are glad to read that the reviewer finds our work to be fitting for publication in Nature Comm., provided that we properly resolve the raised questions. We hope that in the revised manuscript, as well as through the explanations provided below, we fully address all raised questions and the reviewer will recommend publication.

Comment:

1. A general question. Can this square-root topological insulator be extended to higher dimensional systems beside 1D?

Response:

The reviewer raises an interesting question. While we developed a general procedure to analyse square root models, based on the properties of the $SU(3)$ algebra (and its $SU(2)$ subalgebra), we are indeed still limited to 1D. The core idea of our approach is to decompose the Hamiltonian into its basis and use the properties of the underlying algebra to properly define the square operation, and the corresponding topological invariant. Based on this general concept, we are positive that this procedure can be extended to higher dimensions. We are currently working on exactly such an extension, and following the reviewer comment have added a reference to such extensions in the outlook of the main text.

Comment:

And can it be extended to be beyond the tight-binding model?

Response:

The reviewer raises an interesting question. The current model and its analysis are indeed based on a tight-binding description. As demonstrated by the experiment, it is nevertheless still valid for continuous systems, which can, in good approximation, be described within a tight-binding framework. Within the tight-binding framework, we are neglecting, e.g., the coupling to higher modes or radiation modes, which are not part of our theoretical description. In our case, since those effects do not contribute significantly, our theory is still applicable. When these terms become dominant and break the $SU(3)$ structure of the problem, our analysis will no longer hold. We would like to emphasize that our results also reveal interesting properties of generic Hamiltonians (hence also those of continuous models) written on a sub-basis of $SU(3)$. For example, any unitary transformation of the Hamiltonian has no effect on the squared Hamiltonian, hence the topology is expected to be preserved. However, unitary operations span a large group and can lead to various, seemingly unrelated, models. Hence, this construction can be used to engineer Hamiltonians beyond the specific example of the Aharonov-Bohm cages, or even beyond Hermitian models.

Comment:

2. The introduction of this work focus on the meaning of square-root topological insulators and their importance. As pioneer and earlier works along this thought, the contributions of Refs. [37, 38] should be highlighted in the beginning.

Response:

We thank the reviewer for this comment and fully agree that the mentioned contributions were of big importance for our work. Therefore, we now highlight those at the beginning of our revised manuscript.

Comment:

3. At $\phi=0$, three bands cross. This feature cannot be observed in Fig. 1b. I guess a transparent mesh would be better.

Response:

We fully agree and changed the corresponding figure in the revised version of the manuscript.

Comment:

4. The last paragraph on page 2, the authors mentioned that "For a standard TI, the winding phase takes ...". I guess here the authors refer to SSH model. As nowadays, the term TI include many different systems. I think it is better to be more precise here.

Response:

We thank the reviewer for pointing this out. Indeed, the term TI refers to a variety of different manifestations where topological effects appear in various observables and codimensions, where the latter describes boundary effects of different dimensions such as edges or corners of 2D TIs. In 1D, however, one can only analyze the polarization of the chain and care about 0D boundaries. The corresponding examples of 1D TIs are, however, not limited to just the SSH model. The topological classification of 1D insulators includes models with (possibly) nontrivial topology from classes BDI, D, DIII and CII. In this sense, we have kept our discussion more general. At the same time, following the Reviewer's comment, in order to interact with a wider readership, we now specify that we are interested in strong 1D TIs and refer to the specific example of the SSH model.

Comment:

5. The authors adopt the method presented in Ref. [34] to introduce π flux. The authors should also mention that this method also introduce an onsite energy shift besides the negative hopping phase. What is the effect of this onsite energy shift?

Response:

The method used to create the negative hopping (π flux) is based on an auxiliary waveguide with a big onsite energy shift (denoted Δ in section IX of the supplementary material) and smaller onsite energy shifts of the outer waveguides (denoted δ in section IX of the supplementary material). The onsite energy shifts are designed to energetically match the negative coupler with the rest of the lattice. In other words, if this compensation would not be done, the two waveguides, which have an effective negative coupling, would be energetically shifted from the rest of the lattice. Experimentally this corresponds to a small change in the refractive index, which is achieved by changing the writing speed when creating the waveguides. Note that, those small changes have almost no influence on the coupling coefficients. In order to clarify this aspect, we added an additional sentence in the revised version of the manuscript.

Comment:

6. Following this question, can arbitrary value of flux be realized in this system?

Response:

While theoretically the AB-cages model can support an arbitrary flux, the above experimental approach can only be used to realize π flux. There are other methods that are used for generating synthetic gauge fields, but in coupled waveguide arrays it is to date solely possible using either Floquet modulation or using the method proposed in Ref. [30]. We, nevertheless, also identify benefits of the methods, namely with this approach we can be certain to obtain exactly a flux of π , without fluctuation in the synthetic gauge field. We even see this as a feature that can be useful for technological applications, since the case of flux= π corresponds to completely flat bands, which are currently object of intensive research.

Comment:

7. A few typos. Figs. 3 and 4, I can hardly believe the scale bar along the x direction is 50m.

Response:

We apologize for this flaw, which resulted from an erroneous compilation with Latex. We fixed this issue in the revised version of the manuscript, together with minor adaptation of the font sizes etc.

Comment:

8. The authors mentioned that the light field is measured directly through fluorescence microscopy. Does this mean that light is leakage from side.

Response:

The reviewer is correct with the assumption that a small amount of light is leaking out of the waveguide due to fluorescence. This fluorescence is deliberately induced and harvested in order to gain information about the light propagation within the glass chip. Depending on the amount of Oxygen within the material, color centers form during the waveguide fabrication process along the waveguide. It may be noted that the amount of light leaking out of the waveguide is small, compared to the guided light. If this effect is not desired e.g., for technical applications or quantum measurements, one can use glass with a low amount of oxygen such that almost no color centers are formed.

Comment:

9. A control experiment which shows how the wave propagate if the band is not flat would be great in supplement of Fig. 3.

Response:

We thank the reviewer for this suggestion and think that this additional information is useful for illustrating the transformation from dispersive to flat bands. Therefore, we added experimental pictures of edge and bulk excitation with vanishing flux. Both show no signs of localization, in contrast to the flux = π case. The additional figure can be found in the revised supplementary material.

REVIEWERS' COMMENTS:

Reviewer #2 (Remarks to the Author):

The report is well written and has successfully addressed all my concerns raised in the last report. I will recommend this work for publication.

Reviewer #2 (Remarks to the Author):

Comment:

The report is well written and has successfully addressed all my concerns raised in the last report. I will recommend this work for publication.

Response:

We thank the reviewer for his/her time and effort to evaluate the revised version of the manuscript. We are happy to read that he/she now recommends this work for publication.